# Comparative Study of Nanostructured CuSe Semiconductor Synthesized in a Planetary and Vibratory Mill

**DOI:** 10.3390/nano10102038

**Published:** 2020-10-15

**Authors:** Marcela Achimovičová, Matej Baláž, Vladimír Girman, Juraj Kurimský, Jaroslav Briančin, Erika Dutková, Katarína Gáborová

**Affiliations:** 1Institute of Geotechnics, Slovak Academy of Sciences, Watsonova 45, 04001 Košice, Slovakia; balazm@saske.sk (M.B.); briancin@saske.sk (J.B.); dutkova@saske.sk (E.D.); gaborova@saske.sk (K.G.); 2Faculty of Science, Pavol Jozef Šafárik University, Košice, Park Angelinum 9, 04154 Košice, Slovakia; vladimir.girman@upjs.sk; 3Faculty of Electrical Engineering and Informatics, Technical University Košice, Letná 9, 04200 Košice, Slovakia; juraj.kurimsky@tuke.sk

**Keywords:** mechanochemical synthesis, planetary ball mill, industrial vibratory mill, nanostructured semiconductor

## Abstract

Copper(II) selenide, CuSe was prepared from Cu and Se powders in a stoichiometric ratio by a rapid, and convenient one-step mechanochemical synthesis, after 5 and 10 min of milling in a planetary, and an industrial vibratory, mill. The kinetics of the synthesis, and the structural, morphological, optical, and electrical properties of CuSe products prepared in the two types of mill were studied. Their crystal structure, physical properties, and morphology were characterized by X-ray diffraction, specific surface area measurements, particle size distribution, scanning, and transmission electron microscopy. The products crystallized in a hexagonal crystal structure. However, a small amount of orthorhombic phase was also identified. The scanning electron microscopy revealed that both products consist of agglomerated particles of irregular shape, forming clusters with a size ~50 μm. Transmission electron microscopy proved the nanocrystalline character of the CuSe particles. The optical properties were studied using UV–Vis and photoluminescence spectroscopy. The determined band gap energies of 1.6 and 1.8 eV for the planetary- and vibratory-milled product, respectively, were blue-shifted relative to the bulk CuSe. CuSe prepared in the vibratory mill had lower resistivity and higher conductivity, which corresponds to its larger crystallite size in comparison with CuSe prepared in the planetary mill.

## 1. Introduction

Copper (II) selenide CuSe belongs to the group of I-VI transition metal chalcogenides and p-type semiconductors, and has electrical and optical properties that make it an interesting, low-cost, and technologically usable material. CuSe is particularly suitable for photovoltaic applications, namely for solar cells [1,2,3], optical filters [4], film electrodes [5], and has potential as a superionic [6] and rechargeable lithium battery material [7]. A moderate number of scientific publications are devoted to copper selenides research. Over the past two decades, scientists have tried to synthesize CuSe using new methods that do not require high temperatures and poisonous precursors. The hydrothermal method [3,8], sonochemical method [9], solution-phase synthetic route [10], and microwave-assisted method [11] were employed to prepare hexagonal nanoparticles, spherically shaped nanocrystals, flakes, nanoplatelets, and nanosheets of CuSe. In addition, the synthesis of CuSe in the form of thin films was carried out by various deposition techniques, such as the low-temperature dip type method [12], pulsed laser deposition [7], thermal evaporation [13,14], solution growth technique [1], chemical bath deposition [2,15,16,17], electrodeposition technique [18], vacuum evaporation technique [19], and electrochemical/chemical bath deposition [5], namely because of the attractiveness of the application of these thin films in the solar cell industry. Among many available methods, mechanochemical synthesis has an inevitable place, due to its environmentally friendly, solventless, and scalable character [20,21,22]. Ohtani et al. were the only researchers who synthesized γ-CuSe by mechanical alloying, for 60 min using a high-energy ball mill [23].

CuSe exists in three crystallographic modifications, namely α-CuSe (low-temperature hexagonal), β-CuSe (orthorhombic), and γ-CuSe (high-temperature hexagonal) [24]. According to Stolen et al. the first-order α→β transformation occurs at 327 K, and a second-order β→γ transition takes place at 410 K [25].

Several metal selenides, together with binary and ternary copper sulfides were mechanochemically synthesized using a scalable eccentric vibratory mill, and in the case of CuS commercial production was also verified [26,27,28,29,30,31,32]. Different types of mill, producing a different nature and intensity of mechanical energy, which is applied to the mechanochemical synthesis, can determine its course, and the formation of centers responsible for a solid-state reaction [33]. The products of the same mechanochemical reaction obtained using different types of mills may differ in crystal structure, morphology, and properties, as has already been demonstrated in a couple of papers by our research group [27,34]. The planetary and vibratory mills differ in several aspects of the working regime. The planetary ball mill uses the principle of centrifugal acceleration, resulting from a combination of two opposite centrifugal fields and causing a friction effect. The material inside a chamber performs two relative motions: a rotary motion around the mill axis, and a planetary motion around the chamber axis [33]. The eccentric vibratory mill performs inhomogeneous elliptical, circular, and linear vibrations, whereby the amplitude of the individual milling balls and the rotation speed of the ball filling increases, and its resulting direction of movement is determined. According to Heegn, the planetary mill reaches a higher maximal relative acceleration, b/g = 15, than the eccentric vibratory, b/g = 10 [35].

In this paper, we report the preparation of CuSe by rapid one-pot mechanochemical synthesis, in laboratory planetary and industrial eccentric vibratory mills, suitable for large-scale production, and compare the products in terms of structural, morphological, optical, and electrical properties.

## 2. Materials and Methods

Mechanochemical synthesis of CuSe was performed by the milling of copper (>99.7%, 54.02 μm, Pometon GmbH, Germany) and selenium powders (99.5%, 74 μm, Aldrich, Germany) in the planetary ball mill (Pulverisette 6, Fritsch, Germany) and the industrial eccentric vibratory mill ESM 656-0.5 ks (Siebtechnik, Germany) according to the reaction:Cu + Se = CuSe(1)

The reaction is thermodynamically feasible because of the negative value of enthalpy change $\Delta H_{298}^{0}=-41.81\;\mathrm{kJ}.{\mathrm{mol}}^{-1}$ [36]. The following conditions were used for the mechanochemical synthesis in the planetary ball mill: loading of the mill = 50 balls (10 mm in diameter), the material of the milling chamber and balls = WC, the volume of the milling chamber = 250 mL, the mass of Cu powder = 2.23 g, the mass of Se powder = 2.77 g, ball-to-powder ratio = 73:1, milling atmosphere = Ar, rotation speed = 550 rpm, and milling time = 1.5, 3, and 5 min.

Experimental conditions for mechanochemical synthesis in the industrial eccentric vibratory mill were as follows: loading of the satellite chamber = 83 balls (35 mm diameter), the material of the milling chamber and balls = WC, the volume of the milling chamber = 5 L, the mass of Cu powder = 40.25 g, the mass of Se powder = 50 g, ball-to-powder ratio = 337:1, the milling atmosphere = Ar, the frequency of the motor = 960 rpm, the amplitude of the inhomogeneous vibrations = 20 mm, and the milling times = 5 and 10 min.

The products of mechanochemical synthesis were characterized by X-ray diffraction analysis (XRD), the specific surface area measurements (BET), the particle size distributions (PSD), scanning electron microscopy (SEM), energy dispersive X-ray analysis (EDX), and transmission electron microscopy (TEM). The optical properties were evaluated by absorption UV–Vis spectroscopy and emission photoluminescence spectroscopy (PL).

The XRD measurements were carried out in Bragg-Brentano geometry using a D8 Advance diffractometer (Bruker, Germany), working with Cu_Kα_ radiation and the Diffracplus Eva tool, and ICDD-PDF 2 database was utilized. Rietveld refinement for the quantitative phase analysis and crystallite size estimation was performed using TOPAS Academic software [37,38].

The BET measurements were determined by the low-temperature nitrogen adsorption method in a Gemini 2360 sorption apparatus (Micromeritics, USA).

PSD was measured by a laser diffraction system using a Mastersizer 2000E particle size analyser (Malvern Pananalytical, UK), with a dry feeder Scirocco 2000M, and in the measuring range 0.02–2000 μm.

SEM study was performed using a microscope, MIRA3 FE-SEM (TESCAN, Czech Republic), equipped with an EDX detector (Oxford Instrument, Oxford, UK).

TEM observations were carried out using a JEOL 2100F UHR microscope, equipped with a Schottky field emission source and operated at 200kV. The images were taken in high-resolution mode, and for structure identification, the selected area diffraction was used. Regarding electron diffraction experiments, the microscope was precisely calibrated using MoO_3_ crystal, and for double-checking, gold nanoparticles were used as well. Both samples were dispersed in pure ethanol and ultrasonicated for 10 min to reduce the agglomeration effect of crystals prior to the measurement. Drops of the solutions were put on a conventional copper support grid covered with flat carbon film. Subsequently, the samples were stored in a vacuum to eliminate ethanol in the samples.

The absorption spectra were measured using a UV–Vis spectrophotometer, Helios Gamma (Thermo Electron Corporation, UK), in quartz cell, by dispersal of the synthesized particles in absolute ethanol by ultrasonic stirring.

The emission photoluminescence (PL) spectra at room temperature were acquired at the right angle on a photon-counting spectrofluorometer PC1 (ISS, USA), with an excitation wavelength of 480 nm. A 300 W xenon lamp was used as the excitation source. Excitation and emission slit widths were set at 1 and 2 mm. For the measurement, a 1-cm path length rectangular quartz cuvette was used. The PL intensity was measured from the CuSe powders ultrasonically dispersed in absolute ethanol.

The electrical properties of CuSe pellets were studied using a standard four-point probe technique [39,40]. The pellets were individually produced in a laboratory hydraulic pellet/tablet press (Specac, USA), under pressure of 1 t, at room temperature, without retention time, from 0.37 g of CuSe powder products. The measurements were taken with a source-measure unit while the four-point test head (Ossila Ltd., Sheffield, UK) was attached to the sample under test, Figure 1a. The test head was fixed symmetrically to the geometrical center of the sample surface. To obtain a reproducible result, the probe tips were loaded with constant contact force and fixed at the same position, Figure 1b. The CuSe pellets were circular, with a diameter of 7.05 mm, an average thickness of 2.2 mm, and an average room-temperature density 4.31 g × cm^−3^. Probe spacing was 1.27 mm.

## 3. Results and Discussion

The XRD patterns shown in Figure 2 indicate that the mechanochemical reaction between Cu and Se in the planetary mill started after a very short time. After 1.5 min of milling two copper phases, orthorhombic CuSe (ICDD-PDF2 027-0184) and Cu_2_Se_3_ (ICDD-PDF2 03-065-1656) were simultaneously formed. Cu_2_Se_3_ is formed because of its more negative standard enthalpy of formation $\Delta H_{298}^{0}=-124.27\;\mathrm{kJ}.{\mathrm{mol}}^{-1}$ compared to $\Delta H_{298}^{0}=-41.81\;\mathrm{kJ}.{\mathrm{mol}}^{-1}$ for CuSe formation [36]. However, small peaks of unreacted Cu and Se are still seen. Already after 3 min of milling only CuSe, copper selenide (ICDD-PDF2 34-0171) as a major phase was found, and the same situation also persisted after 5 min of treatment.

The XRD patterns in Figure 3 show that the mechanochemical reaction in the vibratory mill was very fast as well. After 5 min of milling, the diffractions corresponding to elemental precursors are not visible anymore and just those of CuSe are present. The situation is similar after 10 min. The stronger intensity of the peak of (006) plane documents anisotropic crystal structure [11].

The final products obtained after 5 and 10 min using planetary and vibratory milling, respectively, were subjected to Rietveld refinement. The main results are summarized in Table 1.

The product was found to be composed mainly of the hexagonal modification of CuSe in both cases, but also a small admixture of orthorhombic modification crystallizing in the space group CmCm was found. Ohtani and co-authors managed to prepare a high-temperature hexagonal γ-CuSe after 60 min of milling in a planetary ball mill, without ruling out the possibility of coexistence of orthorhombic β-CuSe as well [23]. The amount of this admixture was higher for the planetary ball-milled product (its content was around 22%). The observed unit cell parameters are in accordance with those reported in the literature for the clockmannite CuSe phase [24]. The estimated crystallite size is almost three times smaller for the planetary ball milled product. In this case, microstrain was also found to contribute to peak broadening. The observed crystallite size supports more intensive treatment in the case of planetary ball milling.

Higher content of the orthorhombic phase, in the case of planetary ball milling, most probably points to a more pronounced first-order transition of α-CuSe to β-CuSe. This is most probably a result of the fact that the local temperatures in the planetary ball mill are higher than in the case of the vibratory mill, as more pronounced reaction progress has been found when comparing the two types of mills in the past [27,34].

The values for specific surface area and the results of particle size distribution analysis of both CuSe products are listed in Table 2. The product prepared in the planetary mill was 4.8-times finer-grained, and thus had a larger specific surface area than the product prepared in the vibratory mill. This difference might have been related to the specific morphology of CuSe/vibratory mill (VM) grains described later.

The morphology and crystalline character of mechanochemically synthesized CuSe were characterized by SEM and TEM. SEM images of CuSe-PM and CuSe-VM particles synthesized in the planetary and vibratory mills are shown in Figure 4 and Figure 5, respectively. Both products consist of agglomerated particles of irregular shape, which form clusters with a size of about 50 μm. In the VM sample (Figure 5), a layered structure can be observed, as opposed to the PM sample, in which the particles are more or less compact. Layered morphology in the case of VM is probably related to the almost pure hexagonal structure of CuSe and its cleavability in a certain direction. The elemental mapping for both samples in Figure 4b and Figure 5b evidenced the homogeneous distribution of Cu and Se. EDX qualitative analysis shown in Figure 4c and Figure 5c confirmed the Cu:Se atomic ratio of nearly 1:1 for both CuSe products.

The results of the TEM analysis of both CuSe products are shown in Figure 6 and Figure 7, respectively.

The average grain size of CuSe-PM and CuSe-VM was determined to be 22.19 and 31.08 nm, respectively. The grains are highly agglomerated, and interfaces of individual nanocrystals cannot be clearly recognized. Figure 6 and Figure 7 show parts of the aggregates at high magnifications. Since the samples look the same, this imaging mode does not provide any relevant information about the structural differences. However, after precise post-processing, significant differences were found by electron diffraction. Diffraction patterns were subjected to radial integration, and for the final determination of reflection distances correction coefficients were applied. These correction coefficients, which increase the accuracy of diffraction data, result from instrumental diffraction error, and were defined based on the calibration of diffraction patterns. The obtained reflections on the diffraction pattern of sample CuSe-PM (inset of Figure 6) correspond to the orthorhombic structure of the CuSe phase, with lattice parameters a = 0.3948 nm, b = 0.6958 nm, and c = 1.7239 nm. Thus, most probably a region containing orthorhombic modification was analyzed in this case. All visible reflections on the diffraction pattern were checked and identified, but for readability, only a few of them have been labeled in Figure 6. The diffraction pattern of sample CuSe-VM, presented in Figure 7, was evaluated in the same manner. This sample was associated with the distinguishing problem, whether the structure is orthorhombic or hexagonal, since both structures generate reflections on overlapping positions, with very tiny differences. Nonetheless, careful processing of the diffraction data revealed a better coincidence with the calculations for the hexagonal structure of the CuSe phase, with lattice parameters a = 0.3939 nm and c = 1.7215 nm. Particularly, the differences were more highlighted for reflections with higher Miller indices. The selected area diffraction patterns (SAED) show nice rings for CuSe-PM (inset of Figure 6), being a proof of nanocrystalline character, whereas the rings are not complete for CuSe-VM product (inset of Figure 7), so the nanocrystals are larger in this case, which is in accordance with the results of the Rietveld refinement of XRD data.

The optical absorption behavior of the CuSe products mechanochemically synthesized for 5 and 10 min in the planetary and vibratory mills, respectively, was studied by UV–Vis spectroscopy in the region of 200–1100 nm. The UV–Vis spectra of both products in Figure 8 are almost identical and show a broad and increasing absorption in almost the whole measured optical region, with the exception of the range ~600–800 nm, where there is a slight decrease in absorption. No significant band gaps were observed on the spectra. The experimental optical band gap values for both CuSe products were therefore evaluated on the Tauc’s approach [41]. From the plot (αhν)^2^ vs. hν (Figure 9), the energy band gaps of 1.6 eV and 1.8 eV were determined for mechanochemically synthesized CuSe in the planetary and vibratory mill, respectively. This was done by extrapolation of the linear region of the plot to (αhν)^2^ = 0, which indicates a direct optical transition in these semiconductors. The determined values of the band gap energies were blue-shifted relative to the value 1.05 eV for bulk CuSe [42,43], assigned to the optical transitions of the excitonic states in CuSe products, and could be attributed to the existence of very small nanocrystallites in the product.

Table 3 summarizes the published values of CuSe direct band gaps prepared by various methods. The bandgap values of the CuSe prepared by simple, fast, and one-pot mechanochemical synthesis correlate well with the CuSe band gap values prepared by a single source-route, according to [43]. However, in comparison with the values of >2 eV, determined for CuSe prepared by other methods listed in Table 3, our band gaps were red-shifted. In our case, the prepared CuSe products consisted of the nanoparticles agglomerated into the large clusters. This means that band gap energy depends on the crystalline structure size, as well as the contribution of another copper selenide phase. The calculated value of the band gap was close to the optimum for solar cells, indicating that this material could be suitable for solar cell applications.

The optical properties of the mechanochemically synthesized products were also studied by using photoluminescence spectroscopy. PL spectroscopy, as a non-destructive method, gives information about the electronic transitions present in the sample upon irradiation with light of a particular wavelength. The PL spectra of both CuSe, obtained using an exciting wavelength of 480 nm, are presented in Figure 10. The weak emission peak located at 520 nm (2.37 eV) detected for both samples is almost consistent with the literature [19]. The PL intensity for CuSe-PM is higher in comparison with the one prepared in the vibratory mill. It is in good agreement with the UV–Vis spectra (Figure 8). The higher PL intensity of emission peak may be associated with the structure defects, such as vacancies and interstitial ions, and also surface defects [45]. It is assumed that all defects in nanocrystals are generated as a consequence of high-energy milling. With the increasing number of defects in the structure, the number of electrons in the bandgap increases, which leads to the photoluminescence increase. Analogous to other reported CuSe systems, the size and morphology of the different CuSe products may be responsible for the differences in the shift of emission peaks in the spectra [46,47].

The electrical properties of both CuSe products were also investigated. Namely, sheet resistance, resistivity, and conductivity, which were derived from the measurement data population, consisting of 200 values for each CuSe product. The resulting values are shown in Table 4.

The higher conductivity value of the VM product corresponds to its larger crystallite size (around 64 nm), while lower conductivity refers to the PM product, with crystallite size around 24 nm. The formation of large crystallites reduced the grain boundaries, and therefore reduced grain boundary scattering of carriers and improved the conductivity [17]. The electrical resistivity at ambient temperature was of the expectable order. It was in agreement with findings reported for bulk, or chemically deposited, copper selenide thin films [17,48]. The difference in resistivity values between the products was only very slight, which is related to the TEM grain size of the PM (~22.19 nm) and VM (~31.08 nm) products, which were not greatly different. According to Bakonyi and co-authors the relevant structural parameter for electrical resistivity is the grain size determined from TEM images [49].

## 4. Conclusions

Environmentally friendly, solventless, time-saving, and scalable mechanochemical synthesis of nanostructured semiconductor copper selenide (CuSe) in two types of mill, planetary and vibratory, was realized. X-ray diffraction confirmed that 5 and 10 min of milling in a planetary and vibratory ball mill, yields CuSe product with an average crystallite size of 24 and 64 nm, respectively. The observed smaller crystallite size supported more intensive treatment in the case of planetary ball milling. Rietveld refinement of both products revealed the hexagonal structure of CuSe (ICDD-PDF2 34-0171, space group Pmmc/63), with a small admixture of orthorhombic modification (ICDD-PDF2 27-0184, space group CmCm); the content of admixture being higher after the planetary ball milling. The SEM analysis revealed that both products consist of agglomerated nanoparticles of irregular shape, which form clusters with a size of about 50 μm. EDX qualitative analysis confirmed a Cu:Se atomic ratio of nearly 1:1 for both CuSe products. TEM analysis and indexing of the SAED pattern revealed a good fit with the calculations for the hexagonal structure of the CuSe phase. The energy band gaps of 1.6 and 1.8 eV of CuSe synthesized in the planetary and vibratory mill, respectively, were in the range for potential photovoltaic applications. The PL emission spectra of both CuSe products indicated only a weak emission peak at 520 nm. Regarding the electrical properties, CuSe prepared in the vibratory mill had higher conductivity, which corresponds to its larger crystallite size in comparison with CuSe prepared in the planetary ball mill. The electrical resistivity at ambient temperature was of an expectable order, and had almost the same value for both CuSe products. This comparative study contributed to the findings that in mechanochemical synthesis, the structural, morphological, optical, and electrical properties of both CuSe semiconductors, prepared in different types of mill, show only small differences, and that the industrial eccentric vibratory mill can be suitable for large-scale CuSe production.

## Figures and Tables

**Figure 1 nanomaterials-10-02038-f001:**
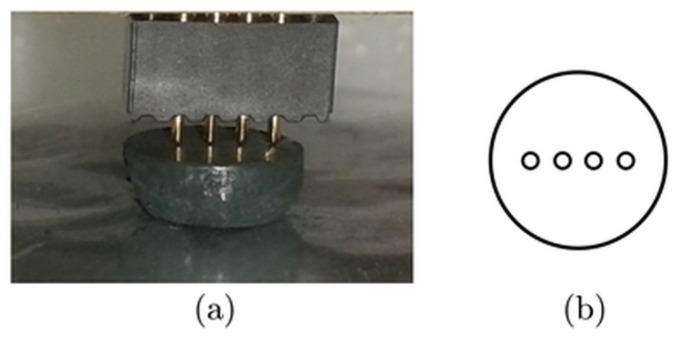
Four-point probe test arrangement: (**a**) Test head on the pellet photo; (**b**) Probe tips position (not to scale).

**Figure 2 nanomaterials-10-02038-f002:**
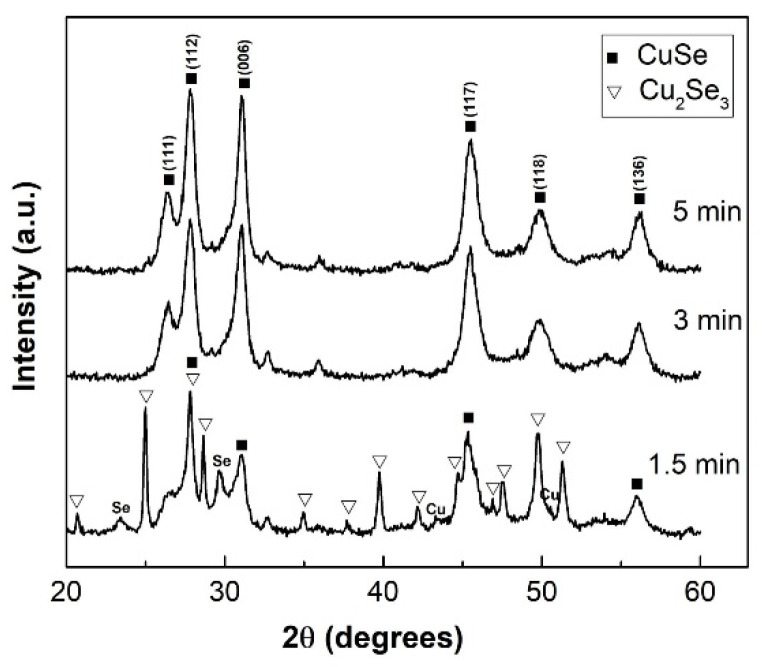
XRD patterns of Cu/Se mixtures milled in the planetary mill for 1.5, 3, and 5 min. Typical reflections are assigned to corresponding phases: CuSe, Cu_2_Se_3_, Se = selenium, Cu = copper.

**Figure 3 nanomaterials-10-02038-f003:**
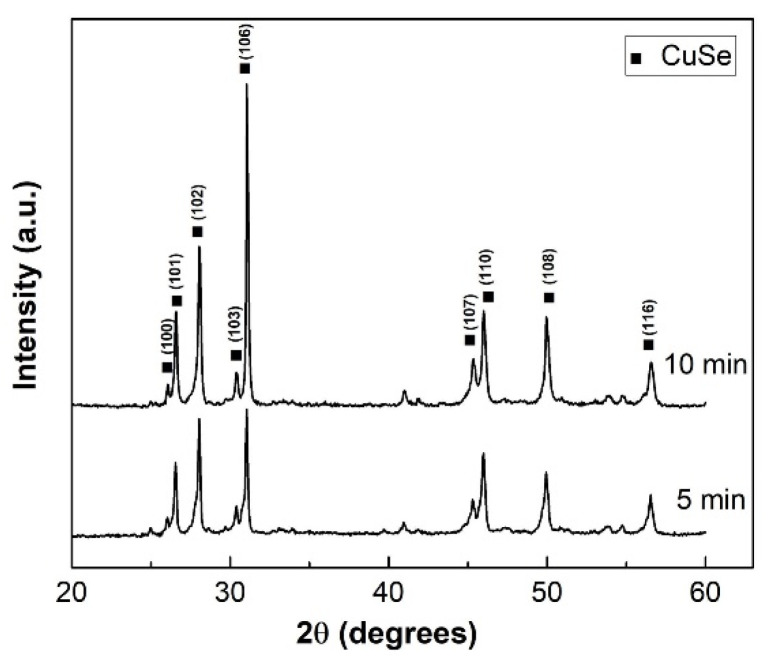
XRD patterns of Cu/Se mixtures milled in the vibratory mill for 5 and 10 min. Typical reflections are assigned to the CuSe phase.

**Figure 4 nanomaterials-10-02038-f004:**
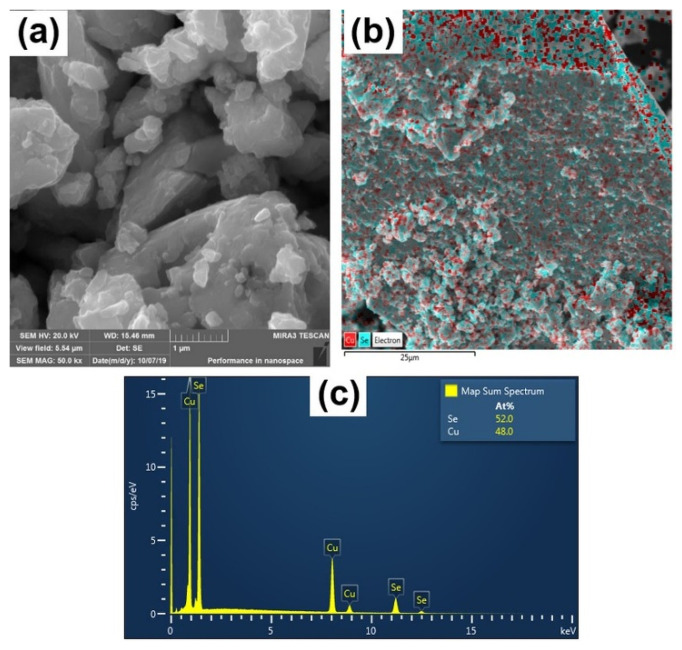
SEM analysis of CuSe-PM: (**a**) image; (**b**) elemental mapping; (**c**) EDX spectrum.

**Figure 5 nanomaterials-10-02038-f005:**
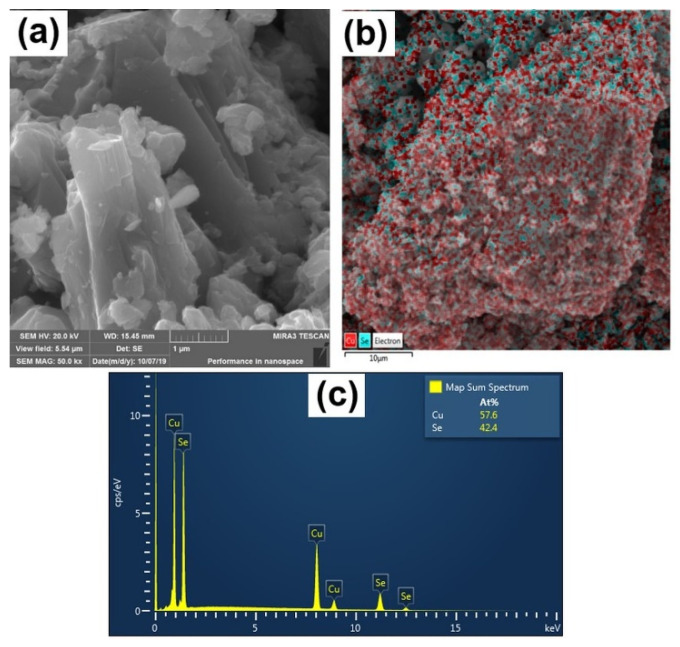
SEM analysis of CuSe-VM: (**a**) image; (**b**) element mapping; (**c**) EDX spectrum.

**Figure 6 nanomaterials-10-02038-f006:**
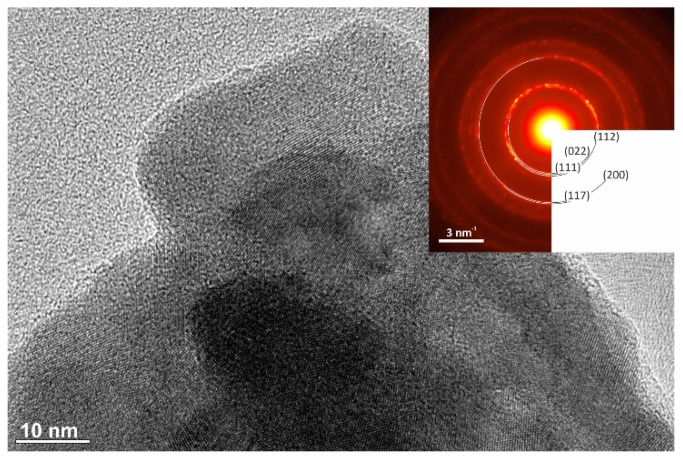
HRTEM image of agglomerated crystals of sample CuSe-PM, imaged at magnification 300 kx, SAED pattern of the same crystals (inset).

**Figure 7 nanomaterials-10-02038-f007:**
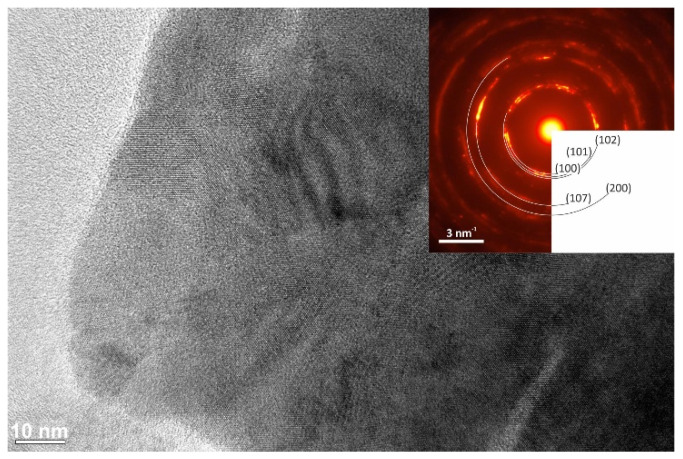
HRTEM image of the structure of the CuSe-VM sample, acquired at magnification 200 kx, corresponding to the SAED pattern (inset).

**Figure 8 nanomaterials-10-02038-f008:**
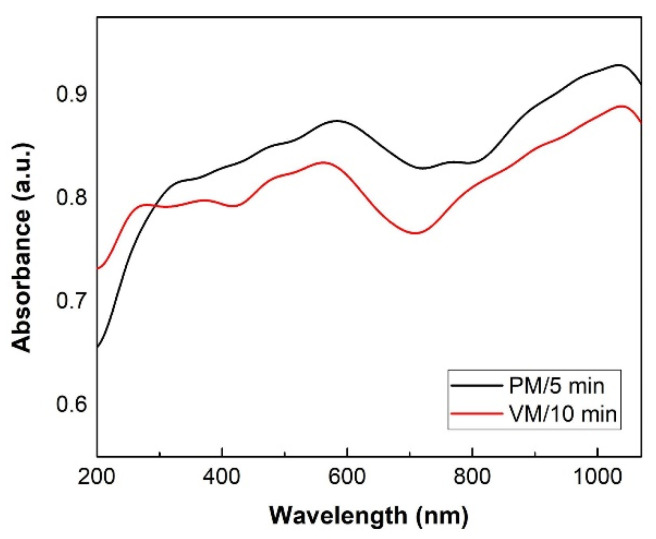
UV–Vis optical absorption spectra of CuSe products after 5 and 10 min of mechanochemical synthesis, in planetary (PM) and vibratory mills (VM).

**Figure 9 nanomaterials-10-02038-f009:**
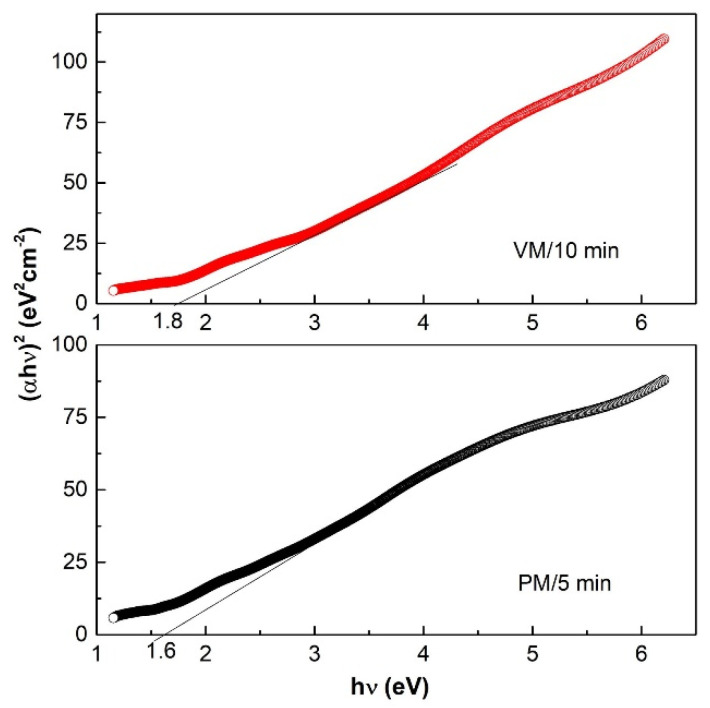
(αhν)^2^ vs. hν plot and the obtained band gap energies, 1.6 and 1.8 eV.

**Figure 10 nanomaterials-10-02038-f010:**
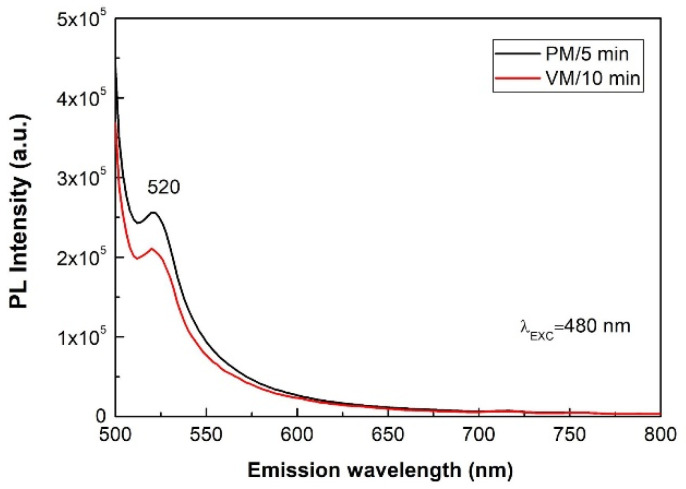
PL spectra of CuSe products after 5 and 10 min of mechanochemical synthesis in planetary (PM) and vibratory (VM) mills at an excitation wavelength of 480 nm.

**Table 1 nanomaterials-10-02038-t001:** Results of the Rietveld refinement of final CuSe products.

	CuSe-PM	CuSe-VM
Amount of hexagonal CuSe (Pmmc/63) (%)	78.2 ± 0.8	98.4 ± 0.7
Crystallite size (nm)	24 ± 2	64 ± 2
Strain (%)	1.45 ± 0.07	not contributing
Unit cell parameter a (nm)	0.3972 ± 0.0017	0.3941 ± 0.0003
Unit cell parameter c (nm)	1.7215 ± 0.0074	1.7232 ± 0.0017

PM-planetary milled; VM-vibratory milled.

**Table 2 nanomaterials-10-02038-t002:** Specific surface area, S_A_, and particle mean diameter, x_50_, of the final CuSe products.

CuSe Product/Milling Time	S_A_ (m^2^ × g^−1^)	x_50_ (μm)
PM/5 min	1.60	9.29
VM/10 min	1.02	44.78

PM-planetary milled; VM-vibratory milled.

**Table 3 nanomaterials-10-02038-t003:** The published values of CuSe direct band gaps prepared by various methods.

Method of Preparation	Direct Band Gap (eV)	Reference
Microwave-assisted synthesis	1.36	[11]
Thermal evaporation	1.54–2.54	[14]
Single source-route	1.63–1.77	[43]
Solution growth technique	2.03	[1]
Electrodeposition	2.10	[18]
Electrodeposition/chemical bad deposition	2.20	[5]
Chemical bad deposition	2.40	[16]
Vacuum evaporation method	~2.70	[19]
Ionic exchange chemical reaction	2.79	[44]
Thermal evaporation technique	3.55	[13]
Mechanochemical synthesis-PM	1.6	[This work]
Mechanochemical synthesis-VM	1.8	[This work]

**Table 4 nanomaterials-10-02038-t004:** Electrical properties of CuSe products synthesized in planetary (PM) and vibratory (VM) mills.

	CuSe Product	Mean	StandardDev	Median
Sheet resistance (Ω/square)	PM/5 min	0.70	0.02	0.70
	VM/10 min	0.46	0.02	0.46
Resistivity (mΩ m)	PM/5 min	1.24	0.04	1.24
	VM/10 min	1.20	0.05	1.20
Conductivity (S/m)	PM/5 min	808.19	27.81	807.75
	VM/10 min	831.80	32.71	831.94

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
