# Peer review of "Comparative Study of Nanostructured CuSe Semiconductor Synthesized in a Planetary and Vibratory Mill"

_nanomaterials, 2020, doi:10.3390/nano10102038_

Round 1

Reviewer 1 Report

Sep. 4, 2020

Review on the paper #Nanomaterials-928216

Title “Comparative Study of Nanostructured Cuse Semiconductor Synthesized In a Planetary and Vibratory Mill”
Authors: Achimovicova et al.

which was submitted for publication to Nanomaterials

This manuscript describes a study of the preparation of the semiconducting CuSe phase by ball milling and the study of the structure and relevant poperties. This a nice manuscript reporting properly about the results obtained. The English is mostly also fine. It is definitely suggested for publication in Nanomaterials.

However, the authors should be given the chance to perform some minor revision to improve the paper before final acceptance. Comments are given below for the revision.

Specific questions and comments:

  1. Lines 50/51: The authors mention that Ohtani et al. were the only researchers to produce CuSe by mechanical alloying. If relevant, please discuss in the Results and Discussion section similarities and differences of the results of that study with respect to the present findings.
  2. In Fig. 3, please indicate which XRD pattern is for 5 min and which for 10 min.
  3. In Table 1, the crystallite size and strain parameters are presented. Please specify by which method these data were derived, e.g., classical or modified Williamson-Hall analysis; for the classical, see e.g., D. Cullity and S.R. Stock, Elements of X‑ray Diffraction (Third Edition, Prentice Hall, Upper Saddle River, New Jersey, U.S:A., 2001; for the latter: T. Ungár, Á. Révész and A. Borbély, Dislocations and grain size in electrodeposited nanocrystalline Ni determined by the modified Williamson-Hall and Warren-Averbach procedures. J. Appl. Cryst. 31 (1999) 554-558 or J. Gubicza, X-ray Line Profile Analysis in Materials Science (IGI-Global, Hershey, PA, USA, 2014); ISBN: 978-1-4666-5852-3. It might also be appropriate to display a comparison of, e.g., the WH plot for the two CuSe samples.
  4. It would be nice to display some not high-resolution TEM images in bright-field and dark-field mode and to analyze the grain size distribution from the dark-field images. This would give important additional and complementary information to the XRD crystallite size results as mentioned also below in connection with resistivity data.
  5. The knowledge of grain size by TEM would be important for another reason. The authors provide the resistivity values for the two CuSe samples and find a slight difference between the two samples produced by PM and VM. However, the difference is very small (1.24 and 1.20 milliohm.cm) and it is not sure that it can be ascribed to the crystallite size difference as explained by the authors. Namely, in a recent paper [Bakonyi et al., Phil. Mag. 99, 1139 (2019)] the grain-size dependence of resistivity was reported for nc-Ni and from the relation found there, one cannot rationalize the small change of resistivity for the two CuSe samples with strongly different crystallite sizes (24 and 64 nm). Although Ni is a metal and CuSe is a semiconductor and the resistivity vs. grain dependence may be different for the two cases, it is noted that in this mentioned paper it was also revealed that displaying the resistiviy against TEM grain size and XRD crystallite size leads to quite different results due to the differences in the meaning of the two microstructural parameters. Please consult this paper for further details.
  6. Concerning the resistivity, there are some more notes. Please compare your resistivity to bulk CuSe data or other available ones (e.g., Ref. 17 in your paper gives fairly similar resistivity values for CuSe films). Also, please specify the way (time, temperature, pressure) you have compacted the pellet used for the resistivity measurements. It would be also important to know the density of your pellet, how close it is to bulk CuSe, revealing if there is any residual porosity in the pellets.

English revision comments (some examples are only given, please check the whole text):

  1. Title: “Cuse” --> “CuSe”
  2. Line 35: “low cost” --> “low-cost”.
  3. Line 199: “the significant differences” --> “siginificant differences; line 212: “reveals the better coincidence” --> “reveals a better coincidence”; please check the whole text for definnite and indefinite articles.
  4. Line 202: “results” --> “result”
  5. Line 262: “the number amount of electrons” --> “the number of electrons”.
  6. Line 263: “others” --> “other”
  7. Lines 167 and 285: “more intensive treatment” --> “more intensive treatment”.
  8. Turn of lines 288/289: “based on the SEM analysis revealed” --> “The SE analysis revealed”.

Reviewer 2 Report

The authors report on the obtention of nanocrystalline CuSe by ball milling. The material is interesting, and the characterization methods used are adequate. However, some of the results are not clear and need a better explanation. For instance:

  • The shape of absorbance curves and Tauc´s plots are weird. The authors should explain in more detail the acquired spectra, they seem to be rather noisy and the linear part that has to be used to extract the optical bandgap is not clearly visible.
  • PL experiments were performed with an excitation wavelength of 480nm and a very large slit width for both excitation and emission. These facts lead to a very broad tail in the spectra which is coming from the lamp that can hide other luminescence emissions from the samples. So, it is desirable to carry out the experiments in other conditions to improve the spectral resolution and to get a better insight on the luminescence properties of the material.
  • Finally, the authors claim the optical bandgap of their samples, which varies between 1.6 and 18. eV), is larger than the bulk bandgap due to quantum confinement effects. Have they calculated the particle size needed to obtain such a large effect (0.6 to 0.8 eV shift)? Is it consistent with the particle size observed in TEM results?

In summary, I do not recommend the publication of the manuscript in the present form. The former points should be clarified in order to improve the manuscript to be suitable for publication in Nanomaterials.

Round 2

Reviewer 2 Report

I thank the authors for their explanations and changes.